# Predicting Maternal and Infant Tetrahydrocannabinol Exposure in Lactating Cannabis Users: A Physiologically Based Pharmacokinetic Modeling Approach

**DOI:** 10.3390/pharmaceutics15102467

**Published:** 2023-10-14

**Authors:** Babajide Shenkoya, Venkata Yellepeddi, Katrina Mark, Mathangi Gopalakrishnan

**Affiliations:** 1Center for Translational Medicine, University of Maryland School of Pharmacy, Baltimore, MD 21201, USA; 2Division of Clinical Pharmacology, Spencer Fox Eccles School of Medicine, University of Utah, Salt Lake City, UT 84112, USA; 3Department of Molecular Pharmaceutics, College of Pharmacy, University of Utah, Salt Lake City, UT 84112, USA; 4Department of Obstetrics, Gynecology and Reproductive Sciences, University of Maryland School of Medicine, 11 S Paca, Suite 400, Baltimore, MD 21042, USA

**Keywords:** PBPK, cannabinoids, weed, marijuana and breastfeeding, pediatric cannabis exposure, cannabinoids in breastmilk, weed and breastfeeding, smoking and lactation, cannabis smoking, THC in breastmilk, infant THC exposure

## Abstract

A knowledge gap exists in infant tetrahydrocannabinol (THC) data to guide breastfeeding recommendations for mothers who use cannabis. In the present study, a paired lactation and infant physiologically based pharmacokinetic (PBPK) model was developed and verified. The verified model was used to simulate one hundred virtual lactating mothers (mean age: 28 years, body weight: 78 kg) who smoked 0.32 g of cannabis containing 14.14% THC, either once or multiple times. The simulated breastfeeding conditions included one-hour post smoking and subsequently every three hours. The mean peak concentration (C_max_) and area under the concentration–time curve (AUC_(0–24 h)_) for breastmilk were higher than in plasma (C_max_: 155 vs. 69.9 ng/mL; AUC_(0–24 h)_: 924.9 vs. 273.4 ng·hr/mL) with a milk-to-plasma AUC ratio of 3.3. The predicted relative infant dose ranged from 0.34% to 0.88% for infants consuming THC-containing breastmilk between birth and 12 months. However, the mother-to-infant plasma AUC_(0–24 h)_ ratio increased up to three-fold (3.4–3.6) with increased maternal cannabis smoking up to six times. Our study demonstrated the successful development and application of a lactation and infant PBPK model for exploring THC exposure in infants, and the results can potentially inform breastfeeding recommendations.

## 1. Introduction

The use of cannabis, including smoking, has become increasingly prevalent worldwide, with pregnant individuals and lactating mothers being no exception [1,2]. The growing legalization of recreational cannabis use is expected to contribute further to this phenomenon [3,4]. However, the health risks associated with cannabis smoking are a matter of concern, particularly for vulnerable populations such as lactating mothers and their breastfed infants. Smoking cannabis during lactation may lead to potential prolactin suppression and alterations in both the volume and nutritional composition of breast milk, potentially impacting the amount and quality of breast milk consumed by breastfed infants [5,6,7]. Studies conducted on animals have indicated that exposure to cannabis during breastfeeding may have impacts on the baby’s neurodevelopmental outcomes, similar to in utero exposure. These effects may include reduced mental development, irregular sleep patterns, and aggressiveness, as well as lack of attention [2,8]. However, some other studies have claimed no significant difference in developmental outcomes between exposed and non-exposed infants [9,10].

Tetrahydrocannabinol (THC) is the most potent psychoactive compound found in cannabis. It is metabolized in the liver by CYP2C9 and CYP3A4 into its more polar forms, 11-OH-THC and 11-COOH-THC [11,12]. THC has been reported to pass into breast milk, raising concerns about potential adverse effects on the developing infant. While data on the pharmacokinetics of THC in human milk are limited, its favorable properties, such as high lipophilicity and low molecular weight, suggest it can pass into breastmilk. On the other hand, little or no THC metabolite levels are found in breast milk, probably due to the polar nature of the metabolites [13,14]. The reported milk-to-plasma ratio of THC varies widely between studies, ranging from 1.8 to 34.6, but most studies consistently agree on a median value of around 7.0 [15,16,17]. The concentration of THC in human milk reported in prior studies is highly variable, making it often difficult to interpret the few studies that reported concentrations in milk and time post-exposure. For example, in a study involving eight lactating women, the observed THC concentrations in milk samples collected 1 h after smoking ranged between 12.2 and 420 ng/mL [13]. These studies often rely on self-reported use by lactating mothers, leading to potential discrepancies in the reported use frequency or strength of cannabis consumed. Additionally, conducting adequately controlled studies on lactating mothers is challenging due to ethical and practical reasons [18].

There is a notable lack of consistent and clear information available in clinical guidelines or from health professionals regarding the use of cannabis during breastfeeding. A 2015 survey of 74 lactation professionals revealed varying recommendations, with 44% individualizing their breastfeeding recommendations based on cannabis use frequency, 41% recommending breastfeeding continuation due to perceived benefits outweighing risks, and the remainder recommending cessation of breastfeeding if cannabis must be used [19]. Despite the prevalence of cannabis use among lactating mothers and few studies assessing breastmilk exposures, there is a notable lack of data on infant THC exposure through breastmilk. The complexity of conducting such studies is evident, as it would involve recruiting lactating mothers who use cannabis, many of whom may be hesitant to participate due to fear of judgment. Furthermore, ethically, investigators cannot recommend breastfeeding solely for the purpose of measuring THC concentration in infants.

The currently available literature reports the milk-to-plasma ratio for THC based on single-time point measurements of plasma and breastmilk concentrations. This approach has its limitations as the ratio is expected to vary at different times on the pharmacokinetic profile. Meanwhile, the area under the concentration–time curve (AUC) ratio method is currently impractical due to the challenges in obtaining a full pharmacokinetic profile of THC in breastmilk. The Atkinson method offers an alternative approach to predict the milk-to-plasma ratio using milk and drug characteristics, but it assumes passive drug diffusion and often overlooks the dynamic nature of milk intake, using a fixed daily milk intake of 150 mL/kg/day [20,21].

To address these limitations and better understand cannabis exposure during lactation, we propose utilizing physiologically based pharmacokinetic (PBPK) modeling. The application of PBPK modeling in studying THC pharmacokinetics offers a unique opportunity to bridge the knowledge gap in cannabis exposure during lactation. PBPK models can leverage existing data on cannabis use, pharmacokinetics, and physiological parameters to tailor models specifically for lactating mothers and their infants. By integrating this approach, we can gain a more comprehensive understanding of the dynamics and potential risks of cannabis exposure in this vulnerable population, which may facilitate informed decision-making.

This study has two primary objectives: (1) To develop a paired lactation and infant PBPK model for THC, which includes an oral inhalation compartment to account for maternal smoking behavior, and (2) To predict the complete pharmacokinetic profile of THC in lactating mothers and the corresponding levels in breastfed infants.

## 2. Materials and Methods

### 2.1. PBPK Model Development Workflow

The PBPK model development started with a base intravenous PBPK model to characterize the disposition of THC in adult individuals. The input parameters were adjusted for females, males, or both to allow for the selection of appropriate weight, BMI, cardiac outputs, and organ volumes as needed. Subsequently, the base model was expanded to include oral inhalation and breast compartment, which were incorporated to account for THC smoke particle deposition, absorption, and transfer of THC into breastmilk. These iterative refinements resulted in the final maternal lactation PBPK model. Additionally, a PBPK model specific to infants was developed by modifying the adult intravenous base model to accommodate their unique anatomical and physiological characteristics [22,23]. An oral absorption component was then integrated to obtain the infant oral PBPK model. The overall workflow of the model development process is illustrated in Figure 1.

### 2.2. PBPK Model Development

#### 2.2.1. General Model Structure

The maternal lactation and infant PBPK models included lungs, gut, liver, kidney, muscle, fat, and brain compartments that are involved in drug absorption, distribution, metabolism, and elimination. Other tissues and organs were grouped together into one compartment called the “rest of the body.” The lung was further divided into four sub-compartments to account for the deposition and absorption of smoke particles via inhalation. Furthermore, a breast compartment and a sub-compartment for milk were incorporated to simulate the transfer of THC into breast milk. The infant PBPK model was linked to the maternal model through breastfeeding, and THC ingested through the breastmilk was absorbed into the infant’s body through the gastrointestinal tract (Figure 2).

#### 2.2.2. Base Model: Intravenous PBPK Model

The transfer of THC into organs and tissues was assumed to be limited by blood flow due to its small size and high lipophilic nature. Cardiac output was determined from the virtual subject’s gender, age, and weight as follows [24]:(1)$$CO=\left(6.963+0.446\cdot isMale-0.037\cdot Age+0.013\cdot Weight\right)\cdot 60,$$
where CO is the cardiac output (L/h), isMale is the gender status (0 for female, 1 for male), Age is in years, and Weight is in kg.

The regional blood flow to other organs in the model was represented as a fraction of the cardiac output, as shown in Table 1. However, organ weights were determined based on various factors, including age, weight, height, gender, or a combination of these parameters (Table 1).

Height, on the other hand, was derived from weight and body mass index (BMI) using [36]
(2)$$Height=\sqrt{\frac{Weight}{BMI}}\;.$$

These parameters were used as inputs within the model to represent a virtual individual. The THC-specific parameters used in the model were obtained from the literature and are summarized in Table 2.

The rate of THC distribution into non-eliminating tissue or organ per time was described using the following equation:(3)$$\frac{dA_{organ\left(non-eliminating\right)}}{dt}=Q_{organ}\cdot \left(C_{arterial}-\frac{C_{organ}}{K_{tp,organ}/BP}\right),$$
where $A_{organ}$, $Q_{organ}$, $C_{arterial}$, $C_{organ}$, $K_{tp,organ}$, and BP, is the amount (ng), blood flow (L/h), concentration in arterial blood (ng/mL), concentration in organ (ng/mL), organ tissue-to-plasma partition coefficient, and blood-to-plasma ratio, respectively.

$K_{tp,organ}$ was calculated using the Berezhkovsky-corrected Poulin and Theil method [31,48], which assumes a homogenous distribution of the unbound drug between tissue lipids and water. This method was chosen due to the physicochemical properties of THC, including high log P, low molecular weight, and neutrality at physiologic pH, which makes it well-suited for this approach [31,49]. The THC fraction unbound in plasma, ${fu}_{p}$, was obtained from a prior study [43]. However, the tissue fraction unbound ${fu}_{t}$ was predicted using
(4)$${fu}_{t}=1/\left(1+\frac{1-{fu}_{p}}{{fu}_{p}}\times R\right),$$
where R is a constant with a value of 0.5 for non-adipose tissues and 0.15 for adipose tissues, as described by Poulin and Theil [31].

In eliminating organs such as the liver, Equation (3) is modified by accounting for drug loss due to hepatic metabolism in Equation (4).
(5)$$\frac{dA_{organ\left(eliminating\right)}}{dt}=Q_{organ}\cdot \left(C_{arterial}-\frac{C_{organ}}{\frac{K_{tp,organ}}{BP}}\right)-{CL}_{H}\cdot C_{organ},$$
where ${CL}_{H}$ is the total hepatic clearance.

The hepatic clearance was calculated using the well-stirred liver model assumption, which accounts for all the intrinsic metabolic clearance processes mediated by specific cytochrome P450 (CYP) isoforms. The mathematical representation of the well-stirred liver model is as follows:(6)$${CL}_{h}=Q_{h}\cdot \left(\frac{{fu}_{b}\cdot \sum {CL}_{int,h}}{Q_{h}+{fu}_{b}\cdot \sum {CL}_{int,h}}\right),$$
where $Q_{h}$, ${fu}_{b}$, and $\sum {CL}_{int,h}$, represents hepatic blood flow (L/h), fraction unbound in blood, and total hepatic intrinsic clearance (L/h), respectively.

Based on the free drug hypothesis, unbound concentrations in both blood and plasma were assumed to be equal at steady state. This allowed for the calculation of fractions unbound in blood, ${fu}_{b}$, by dividing ${fu}_{p}$ by blood-to-plasma ratio. The intrinsic clearance of a compound by CYP enzymes can be obtained from data obtained during in vitro metabolic studies using
(7)$${CL}_{int,h\left(CYP\right)}=\left(\frac{V_{\max\left(CYP\right)}}{K_{m\left(CYP\right)}}\right)\cdot \frac{MMPGL\cdot W_{liver}}{{fu}_{mic}},$$
where ${CL}_{int,h\left(CYP\right)}$, $V_{\max\left(CYP\right)}$, and $K_{m\left(CYP\right)}$ represent the intrinsic clearance of a compound by a particular CYP enzyme, the maximum metabolism rate of that enzyme, and the Michaelis–Menten constant for that enzyme, respectively. MMPGL, $W_{liver}$, and ${fu}_{mic}$ represent the mass of microsomal protein per gram of liver, weight of liver, and unbound fraction of drug in microsomes.

The intrinsic clearance of hepatic CYP2C9 and CYP3A4 was used to obtain the total hepatic intrinsic clearance because THC is primarily metabolized by CYP2C9 and CYP3A4 [43,50,51].

#### 2.2.3. Base Model Expansion: Inhalation PBPK Model

In our inhalation model, we added an absorption compartment via oral inhalation to the base model. This compartment represents the respiratory airways and is composed of four distinct sub-compartments: extrathoracic (ET2—mouth, pharynx, and larynx), bronchial (BB—trachea, main bronchi, and intrapulmonary bronchi), bronchiolar (bb—bronchioles from Generations 9 to 15), and alveolar (AL—respiratory bronchioles, alveolar ducts, and sacs from Generations 16 to 26). This approach, which has been previously published in the literature, is based on the human respiratory tract model established by the International Commission on Radiological Protection (ICRP) [26,52].

Cannabis smoke was considered a small particulate matter capable of depositing on the lumen of any of the previously described respiratory airway regions. Regional deposition of smoke particles was calculated using previously described equations in the literature [26,52]. Briefly, each region was conceptualized as a filter, allowing the deposition of smoke particles. Considering the bidirectional airflow (inhalation or exhalation), ET_2_, BB, and bb were assigned two filters each. However, AL was assigned a single filter because the airway is closed at the alveolar end (Figure 3a). The fraction of smoke particles deposited at each filter relied on the tidal airflow through each airway region, as well as the aerodynamic and thermodynamic properties of the particles. Within a particular filter, the transit of particles from prior filters contributed to the addition or depletion of particles in that filter, which affected the overall particles available for dissolution in the epithelial fluid. The dissolved particles were further affected by mucociliary clearance, which facilitated the movement of particles to and away from each airway region. All these processes collectively determined the amount of particles that eventually permeated the lung epithelial cells. A comprehensive depiction of this mechanism is shown in Figure 3b.

The amount of THC present in the lumen, epithelial fluid, epithelium, and intracellular fluid within an airway region at a given time was determined using the following equations:(8)$$\frac{dA_{lu\left(n\right)}}{dt}=k_{{tr}_{n+1}}\cdot A_{lu\left(n+1\right)}-k_{{tr}_{n-1}}\cdot A_{lu\left(n-1\right)}-k_{diss}\cdot \left(S_{thc}-C_{dis\left(n\right)}\right),$$
(9)$$\frac{dA_{elf\left(n\right)}}{dt}=k_{diss}\cdot \left(S_{thc}-C_{diss\left(n\right)}\right)+k_{{mcc}_{n+1}}\cdot A_{elf\left(n+1\right)}-k_{{mcc}_{n-1}}\cdot A_{elf\left(n-1\right)}-f_{un,elf}\cdot {PSA}_{n}\cdot C_{elf\left(n\right)},$$
(10)$$\frac{dA_{ep\left(n\right)}}{dt}=f_{un,elf}\cdot {PSA}_{n}\cdot C_{elf\left(n\right)}-f_{un,ep}\cdot {PSA}_{n}\cdot C_{ep\left(n\right)},\;and$$
(11)$$\frac{dA_{int\left(n\right)}}{dt}=f_{un,ep}\cdot {PSA}_{n}\cdot C_{ep\left(n\right)}-f_{un,int}\cdot {PSA}_{n}\cdot C_{int\left(n\right)},$$
where $A_{lu}$, $A_{elf}$, $A_{ep}$, and $A_{int}$ represent the amount (ng) of THC in the lumen, epithelial fluid, epithelium, and interstitial fluid, respectively; $k_{tr}$, $k_{diss}$, $k_{mcc}$, $S_{thc}$, and $C_{diss}$ represent the transit rate constant (1/h), dissolution rate constant (L/h), mucociliary constant (1/h), the solubility of THC (mg/L), and dissolved concentration in epithelial fluid (mg/L), respectively, $f_{un,elf}$, $f_{un,ep}$, and $f_{un,int}$ represent the fraction unionized in epithelial fluid, epithelium, and intracellular fluid, respectively; PSA, permeability surface area product (L/h); $C_{elf}$, $C_{ep}$, $C_{int}$, concentration (mg/L) in epithelial fluid, epithelium, and intracellular fluid, respectively; n, airway region.

The transit rate constant was obtained by taking the inverse of residence time in each airway region [26], while the mucociliary clearance was calculated using the method described by Hartung and Borghardt [53]. The dissolution rate constant was derived from the diffusion coefficient of THC, the surface area, and the thickness of the epithelial fluid. This calculation was based on the methodologies described by the ICRP and Hintz and Johnson (1989) [26,54]. The permeability surface area product was calculated by multiplying the apparent permeability of lung epithelial cells and the surface area of each region [52,55].

At time *t* = 0, the initial undissolved amount of THC in the airway lumen is calculated using Equation (25).
(12)$$A_{u\left(n\right)t=0}=f_{DE\left(n\right)}\cdot A_{DOSE},$$
where $f_{DE\left(n\right)}$, fraction deposited in nth airway compartment; $A_{DOSE}$, dose inhaled.

The transfer of THC from the alveolar interstitial fluid was assumed to be perfusion-limited due to the abundant supply of blood capillaries and high blood flow in the alveoli. Therefore, the alveolar mass balance equation is
(13)$$\frac{dA_{int\left(AL\right)}}{dt}=f_{un,ep}\cdot {PSA}_{\left(AL\right)}\cdot C_{ep\left(AL\right)}+Q_{AL}\cdot C_{venous}-Q_{AL}\cdot C_{int\left(AL\right)},$$
where $C_{venous}$ is the concentration of THC in venous blood, $Q_{AL}$ is the blood flow to the alveolar, which was assumed to be equal to the cardiac output.

#### 2.2.4. Base Model Expansion: Lactation PBPK Model

To expand our oral inhalation PBPK model, we incorporated a breast compartment that includes a sub-compartment specifically for breast milk, as shown in Figure 2. Considering the processes involved in the secretion and excretion of nutrients into human milk, including exocytosis from alveolar secretory vesicles, coalition of lipids and triglycerides, free permeation, and endocytosis of large immunoglobulins and proteins [56], we integrated this knowledge with the physicochemical properties of THC and existing human milk models [57,58]. As a result, we proposed two potential mechanisms for THC transfer into breastmilk: (1) excretion and reabsorption and (2) permeation. Equations (14) and (15) describe the rate of change of THC in breast tissues and milk.
(14)$$\begin{array}{c} \frac{dA_{breast}}{dt}=Q_{breast}\cdot \left(C_{arterial}-\frac{C_{breast}}{K_{tp,breast}/BP}\right)-{fu}_{p}\cdot f_{un,bt}\cdot {CL}_{btmk}\cdot C_{breast}\\ -f_{un,bt}\cdot {PSA}_{bt}\cdot C_{breast}+{fu}_{mk}\cdot f_{un,mk}\cdot {CL}_{mkbt}\cdot C_{milk}+{fu}_{mk}\\ \cdot f_{un,mk}\cdot {PSA}_{bt}\cdot C_{milk}, \end{array}$$
(15)$$\begin{array}{c} \frac{dA_{milk}}{dt}={fu}_{p}\cdot f_{un,bt}\cdot {CL}_{btmk}\cdot C_{breast}+f_{un,bt}\cdot {PSA}_{bt}\cdot C_{breast}-{fu}_{mk}\cdot f_{un,mk}\\ \cdot {CL}_{mkbt}\cdot C_{milk}-{fu}_{mk}\cdot f_{un,mk}\cdot {PSA}_{bt}\cdot C_{milk}, \end{array}$$
where $A_{breast}$ and $A_{milk}$ is the amount (ng) in breast and milk, respectively; $Q_{breast}$ is blood flow to the breast (L/h); $C_{breast}$ and $C_{milk}$, concentration (ng/mL) in breast and milk, respectively; $K_{tp,breast}$, breast tissue-to-plasma partition coefficient; ${CL}_{btmk}$, clearance (L/h) from the breast into milk; ${CL}_{mkbt}$, clearance from milk into the breast; $f_{un,bt}$, $f_{un,mk}$, fraction unionized in breast and milk, respectively; ${fu}_{p}$, ${fu}_{t}$, ${fu}_{mk}$, fraction unbound in plasma, tissue, and milk, respectively; ${PSA}_{bt}$, permeability surface area product of breast glandular tissue.

A volume of 0.5 L and 0.21 ± 0.011 L was assigned to the breast and breastmilk storage capacity, respectively, based on previously reported data [26,59]. Breast blood flow was assumed to be similar during pregnancy and lactation; hence, we used the reported breast blood flow of 3.5% of the total cardiac output reported during pregnancy [26]. Given the high and variable percentage of fatty tissues in women’s breasts [26,60,61], we did not calculate the partition coefficient of THC in the breast tissues using the Berezhkovsky-corrected Poulin and Theil method. Instead, we assumed $K_{tp,breast}$ to be a fraction of 0.5 (ranging from 0.25 to 0.7) of $K_{tp,adipose}$. This approach was adopted to account for the glandular and fat composition of breast tissues, allowing for a more accurate representation of THC distribution within the breast.

The clearance of THC between breast tissue plasma and milk was predicted using Equations (16) and (17), which were derived based on THC’s physicochemical properties, such as lipophilicity, molecular weight, polar surface area, and hydrogen bond donor properties [58].
(16)$${CL}_{btmk}=10^{-3.912-0.015\cdot PSA+3.367\cdot \log MW-0.164\cdot \left(\log P-\log D_{7.4}\right)}/1000,$$
(17)$${CL}_{mkbt}=10^{2.793+0.179\cdot \log P-0.132\cdot HBD}/1000,$$
where PSA, MW, log⁡P, $\log D_{7.4}$, and HBD represent the polar surface area, molecular weight, logarithm of octanol-water partition coefficient, logarithm of octanol-water distribution coefficient at pH of 7.4, and hydrogen bond donor, respectively.

While the fraction unbound of THC in plasma has been reported in the literature, the fraction unbound of THC, specifically in breast milk, has not been studied. To address this, we utilized the established methodology proposed by Atkinson and Begg (1990) in Equations (18)–(22) to predict the fraction unbound of THC in breast milk [21,58,62].
(18)$${fu}_{mk}=\frac{1}{\frac{f_{w_{skmilk}}}{{fu}_{skmilk}}+f_{f_{milk}}\cdot \log P_{milk}},$$
(19)$${fu}_{skmilk}=\frac{{{fu}_{p}}^{0.448}}{0.000694^{0.448}+{{fu}_{p}}^{0.448}},$$
(20)$$\log P_{milk}=-0.88+1.29\cdot \log P,$$
(21)$$f_{un,bt}=\frac{1}{1+10^{\left(pKa-{pH}_{bt}\right)}},$$
(22)$$f_{un,mk}=\frac{1}{1+10^{\left(pKa-{pH}_{mk}\right)}},$$
where $f_{w_{skmilk}}$ and $f_{f_{milk}}$ are fractional volumes of water and fat in skimmed milk and milk, respectively, which were reported as 0.955 and 0.045, respectively; ${fu}_{skmilk}$ is fraction unbound in skimmed milk; $\log P_{milk}$ is the logarithm of the octanol-water partition coefficient in milk; ${pH}_{bt}$ is the pH of breast tissues, which was assumed to be 7.4; ${pH}_{mk}$ is the pH of breast milk, which was fixed as 7.0, the average reported value between 1 and 10 months postpartum [63,64,65,66].

Due to the lack of data on the permeability of the epithelial cells of the mammary glands, the permeability surface area of the mammary gland was parameterized as follows:(23)$${PSA}_{btgt}=\frac{{SA}_{bt}\cdot D}{h_{bt}},$$
where ${SA}_{bt}$ is the surface area of mammary glands, which was estimated by assuming that each lobule comprising the mammary gland is spherical, and there are 40 lobules per lobe and a total of 20 lobes [67,68]; D is the diffusion coefficient of THC, calculated following ICRP method [26]; $h_{bt}$ is the thickness of mammary epithelial cells, which was fixed as 20 μm [69].

#### 2.2.5. Base Model Reduction: Infant Oral PBPK Model

Age-dependent changes in body weight, height, organ weights, and cardiac output were accounted for by scaling down relevant physiological and biochemical parameters from the base model to infants up to one year of age (see Equations (24) and (25)) [70,71].
(24)$$Weight\left(kg\right)=\frac{\left(Age\left(years\right)\cdot 12\right)+9}{2},$$
(25)$$y=\frac{a+b\cdot age}{1+c\cdot Age\left(years\right)+d\cdot {Age\left(years\right)}^{2}},$$
where y represents physiological parameters such as height (cm), organ weight (kg), or cardiac output (L/min); a, b, c, and d are constant terms, reported in Chang et al. study [71], for each physiological parameter and organ of interest. The blood flow to various organs was considered as a fraction of the total cardiac output. It was assumed that the fractional blood flow to organs in both adults and children was equal.

Alteration in plasma protein concentration and hematocrit level in pediatric populations affects protein binding and blood-to-plasma ratio. Moreso, THC exhibits a strong binding affinity to serum albumin [72,73]. Therefore, the infant fraction unbound and blood-to-plasma ratio of THC were predicted using the following equations [22,74]:(26)$$C_{alb}=1.1287\cdot \log Age\left(years\right)+33.746.$$
(27)$${fu}_{p,child}=\frac{1}{1+\frac{\left(1-{fu}_{p}\right)}{{fu}_{p}}\cdot \frac{C_{alb,child}}{C_{alb,adult}}}.$$
(28)$$K_{p\left(RBC\right)}=\frac{BP-1+H_{t}}{H_{t}\cdot {fu}_{p}}.$$
(29)$${BP}_{child}=1+\left(H_{t,child}\cdot \left({fu}_{p,child}\cdot K_{p\left(RBC\right)}-1\right)\right).$$
where $C_{alb}$ and $C_{alb,child}$ represent the serum albumin concentration of adults and children, respectively; $K_{p\left(RBC\right)}$ is the partition coefficient of unbound THC in red blood cells; $H_{t}$ and $H_{t,child}$ represent the hematocrit level in adults and children, respectively; BP and ${BP}_{child}$ represent the blood-to-plasma ratio in adults and children, respectively.

To account for the ontogeny of CYPs 2C9 and 3A4, which are the major enzymes responsible for the metabolism of THC, their activity was expressed as a fraction relative to adult activity. This was accomplished using Equations (30) and (31), as previously described [71]. These equations allowed for the adjustment of enzyme activity based on age, reflecting the developmental changes in the metabolic capacity of CYPs 2C9 and 3A4 during infancy.
(30)$${CL}_{int,h\left(CYP\right),child}=f_{CYP}\cdot {CL}_{int,h\left(CYP\right),adult}.$$
(31)$$f_{CYP}=\frac{x\cdot {Age\left(years\right)}^{n}}{y+{Age\left(years\right)}^{n}}+z.$$
where ${CL}_{int,h\left(CYP\right),child}$ and ${CL}_{int,h\left(CYP\right),adult}$ are the intrinsic clearance of enzyme (L/h) in adults and children, respectively; $f_{CYP}$ is the enzyme’s fractional activity relative to adult activity; x, y, z, and n are constant terms that are reported in the Chang et al. study [71] for hepatic CYP enzymes.

In the infant model, THC was absorbed from the ingested breast milk via the gastrointestinal tract. The oral absorption process was characterized by two parameters: bioavailability, F, and absorption rate constant, $k_{a}$ (see Equations (32) and (33)).
(32)$$F=F_{a}\cdot F_{g}\cdot F_{h}.$$
(33)$$k_{a}=\frac{2\cdot P_{eff}}{R}.$$
where $F_{a}$, $F_{g}$, and $F_{h}$ represent fraction absorbed, fraction escaping gut metabolism, and fraction escaping hepatic first-pass metabolism, respectively; $P_{eff}$, Caco-2 intestinal permeability; R, the radius of the small intestine. The calculation of each of these parameters has been extensively described elsewhere [75,76].

The dose input in the infant model depended on the maternal lactation model. This involved multiplying the maternal breast milk concentration at each feeding time with the volume of milk the baby ingests during that breastfeeding (Equation (34)).
(34)$${Dose}_{infant}=C_{milk}\cdot V_{milk}.$$
where $C_{milk}$ is the concentration of THC in breastmilk at feeding; $V_{milk}$ is the volume of breast milk ingested by the baby during the feeding session.

The daily volume of breast milk ingested by the baby was calculated using a regression equation derived from a comprehensive analysis of 167 breast milk studies [77]. This equation, generated from the observed data, allows for the estimation of the daily breast milk intake based on the age of the baby (Equation (35)).
(35)$$V_{milk\left(daily\right)}=52.0208-1.9296\cdot Age\left(days\right)+192.5057\cdot \log\left(Age\left(days\right)\right).$$

Based on the guidelines provided by the Center for Disease Control and Prevention and the World Health Organization, exclusively breastfed infants typically feed approximately 8 to 12 times per day [78,79]. To estimate the average volume of breastmilk ingested per feed, it was assumed that the baby feeds every three hours.

#### 2.2.6. Model Training, Verification, and Simulation

To gather pharmacokinetic data on THC in humans following intravenous, inhalation, and oral administration routes, an extensive search of the PubMed and Cochrane literature databases was conducted (Appendix A). Relevant studies were identified using keywords such as “THC pharmacokinetics,” “tetrahydrocannabinol,” “THC inhalation,” “THC milk-to-plasma ratio,” and “cannabis and lactation.” When multiple studies were available, they were divided into two sets: one for model training and the other for model verification. This iterative process was performed sequentially for each step of model development, starting with the intravenous, inhalation, and lactation PBPK models. To replicate the observed concentration profiles from the identified studies, the reported THC profiles were extracted using Plot Digitizer 2.6.6 software (Free Software Foundation, Boston, MA, USA). The performance of the models was evaluated through visual inspection of the pharmacokinetic profiles and the calculation of the absolute average fold error (AAFE) (Equation (36)) for exposure metrics such as AUC or C_max_. An acceptance criterion of AAFE between 1 and 2 was set to establish model validation [80,81].
(36)$$AAFE=10^{\frac{1}{n}\sum \left|\log\left(\frac{Predicted}{Observed}\right)\right|}.$$

Due to the limited or no availability of THC studies in pediatric populations, we proposed an alternative approach for model validation. Initially, the pharmacokinetic profile of oral THC was simulated using reported population pharmacokinetic parameters from a prior oral study. Subsequently, the typical population parameters were scaled down allometrically, taking into account differences in body size and enzyme maturation between adults and pediatric populations [82]. The resulting pediatric population model served as a comparators for verification purposes, enabling the evaluation of the performance and predictive capabilities of our pediatric PBPK model.

Given the significant variability in the reported frequency of cannabis use among both frequent and infrequent smokers, we adopted a range of smoking scenarios to encompass different usage patterns. These scenarios included smoking frequencies of one, two, three, four, five, and six times daily, covering a spectrum from chronic to frequent use. The impact of these smoking scenarios on THC concentrations in breastmilk was evaluated separately at three-hour breastfeeding intervals, beginning at one-hour post-smoking. Breastmilk concentration and volume of milk intake per breastfeeding session were combined to arrive at the dose used to assess the potential exposure of infants to THC. The PBPK model building, verification, and simulation of different scenarios were performed using Pumas^®^ (version 2.3.2, Pumas-AI, Inc., Centreville, VA, USA).

#### 2.2.7. Sensitivity Analysis

A sensitivity analysis was performed on specific input parameters in the PBPK model to assess their impact on the model’s output. The analysis was conducted using the following equation:(37)$$S=\frac{P}{∆P}\cdot \frac{∆PK}{PK}.$$
where S is the sensitivity of the input parameter on PK output; P is the input parameter; ∆P is the change in input parameter; PK is the output PK parameter; ∆PK, change in output PK parameter (AUC_(0–t)_).

Parameters exhibiting sensitivity within the range of −10≥S≥10 were identified as influential on the desired model output.

## 3. Results

### 3.1. Observed Data for PBPK Model Development

Table 3 provides detailed information on the design, participant demographics, and dosing characteristics of the studies used for the model training and verification. It is important to note that two studies were excluded due to potential bioanalytical recovery issues associated with radiolabeled THC [83,84].

For the model training, datasets with a double-blind study design were utilized, while the remaining datasets were used for model verification. However, the limited availability of breastmilk studies providing detailed information on dose, concentrations, and time post-exposure necessitated the use of the same dataset for both training and verification. In the Bertrand et al. study, we assumed a standard THC dose of 14.14% per joint due to the absence of dose information. We included this dataset because it provided THC breastmilk concentrations over an extended post-cannabis use period.

### 3.2. PBPK Model Training and Verification

#### 3.2.1. Intravenous PBPK Model

The anatomical and physiological parameters specific to the system, such as organ weights and regional blood flows in the PBPK model, were consistent with values reported in the literature. The average volume of distribution at steady state was calculated to be 8.44 L/kg, which falls within the range previously reported (6.5–10 L/kg) [43,99,100,101]. The hepatic enzyme kinetics of THC, specifically Vmax and Km for CYPs 2C9 and 3A4, were obtained from a previous study [44] to calculate the intrinsic clearance. Renal clearance was assumed to be negligible due to the low fraction of THC excreted unchanged in urine (<2%) [102,103].

In the simulation, the demographic characteristics and dosage regimens of patients in each study were replicated, but the sample size of virtual subjects was increased to 100 to better capture variability in the simulated profiles. The PBPK model successfully predicted the observed data in the training dataset, and the subsequent model predictions of the remaining datasets were mostly within the stipulated success criteria (Figure 4 and Figure 5).

Since there is limited information available on the differences in THC disposition between chronic and casual cannabis users, these distinctions could not be accounted for in the PBPK model at the time of development. One possible explanation for the simulated AUC not meeting the success criteria for chronic users in the datasets by Kelly and Jones could be enzyme induction associated with smoking behavior, although it was not investigated. Additionally, the lower Cmax reported in the dataset by Meyer et al. compared to the predicted Cmax could be attributed to the injected solution containing both THC and CBD in a 1:1 ratio, potentially leading to interactions between THC and CBD [104].

#### 3.2.2. Inhalation PBPK Model

In the inhalation studies, the dose of THC inhaled was calculated by subtracting the THC content of the original joint from the reported amount remaining in the joint residue. In cases where the residue was not determined, we assumed that the entire THC content of the joint was inhaled. Since only a few studies provided information on smoking topography, we adopted a standard smoking topography of 13 puffs for a standard joint cigarette, with a puff duration of 5 s and an inter-puff interval of 35 s, based on previous reports [105,106] when detailed smoking information was not available. The estimated deposition percentages of the inhaled dose at the ET_2_, BB, bb, and AL airway regions were 0.76%, 1.4%, 37%, and 24%, respectively. Although there were no prior in vivo THC deposition studies for comparison, these values were similar to those reported previously for nicotine [52].

Due to the limited availability of smoking topography data in the studies, only the area under the curve (AUC) was used as the basis for the success criteria. Cannabis inhalation studies that employed a randomized, double-blind design were included in the training dataset to ensure reliability. Overall, the inhalation model development was successful, as over 80% of the observed values in the verification datasets fell within a two-fold deviation of the simulated values (Figure 6 and Figure 7).

#### 3.2.3. Lactation PBPK Model

The development of the lactation PBPK model was successful, with model predictions demonstrating an acceptable absolute average fold error (AAFE) of 1–2 when compared to the observed values (Table 4).

Model verification was conducted using the Baker et al. dataset, while the Bertrand et al. dataset was utilized for visual inspection of the results. The Bertrand study did not report pharmacokinetic parameters, likely due to various factors such as the infrequent collection of breastmilk samples (46 out of 50 participants contributed samples once), the four-year study duration, and self-reported timing of cannabis use. The observed breastmilk concentrations exhibited high variability across the two studies (Figure 8).

This variability may be attributed to ethical challenges in conducting lactation trials, which often rely on self-reporting of cannabis use and mothers collecting their breastmilk samples at their own convenience. Furthermore, the Bertrand study reported that only 64% of participants used cannabis exclusively via inhalation, and the specific method of inhalation was not reported.

Figure 9 shows the simulated THC plasma and breastmilk profiles of a lactating mother who smoked 0.32 g of cannabis containing 14.14% THC once, twice, thrice, four times, five times, or six times daily. Our simulations further showed that THC exhibits a milk-to-plasma AUC ratio greater than 1, indicating higher concentrations in breastmilk compared to plasma, which is similar to previously reported value [16,17]. Additionally, the concentration of THC in breast milk increases with an increasing number of daily smoking frequency.

#### 3.2.4. Infant Oral PBPK Model

The age-dependent physiological and anatomical parameters within the validated intravenous PBPK model were successfully scaled down to account for infants up to one year of age, employing established equations or reported values. This adaptation was particularly relevant, given that infants are typically breastfed for a duration of 6 months to 1 year [107]. Unfortunately, due to the limited availability of infant plasma THC data specifically for infants, conventional training and verification were not feasible. To address this challenge, an alternative empirical approach was adopted, leveraging an existing oral population pharmacokinetic (popPK) model described by Klumpers et al. [108]. In their study, a cohort of 21 subjects (52% female) was administered various doses of oral THC (5 mg, 6.5 mg, or 8 mg). The reported population pharmacokinetic parameters from the study were used to replicate an adult THC popPK model within the Pumas^®^ software (version 2.3.2, Pumas-AI, Inc., Centreville, VA, USA). By reproducing the results from the Klumpers et al. study, we successfully established an adult oral pharmacokinetic model for THC.

To account for age-related differences in drug clearance and volume parameters, we applied allometric scaling based on body weight using the standard allometric scaling equation [109]. However, considering that drug clearance in neonates and infants is additionally influenced by enzyme maturation [110], we incorporated a modified version of the previously established method for morphine in neonates [82] to address this. To verify our infant PBPK model, we administered the previously estimated infant dose of 7.3 mcg/kg (as seen in Table 4) in both the popPK and the PBPK model and simulated plasma concentrations over a 12 h period (Figure 10).

The infant popPK and PBPK model predictions for AUC (neonates—123 vs. 456 ng·hr/mL) and Cmax (neonates—123 vs. 456 ng/mL) were within the acceptable range of 1–2 folds difference for all age groups up to one year. This successful verification supports the reliability and accuracy of the developed infant PBPK model. The simulated THC concentration profiles across the various months, including neonatal periods up to one year of age, exhibited comparable patterns. This observation may be attributed to the substantial contribution of CYP2C9 in THC metabolism [12], as newborns possess approximately 21% of the adult activity of CYP2C9 at birth. Moreover, the expression of CYP2C9 in newborns rapidly reaches adult levels within the first two weeks of life [74].

#### 3.2.5. Simulations

Firstly, within the lactation PBPK model, we obtained THC concentrations in breast milk at three-hour intervals starting from the first hour, which corresponded to each breastfeeding session. This process was repeated for different frequencies of cannabis use that were investigated, as shown in the results in Table 5.

Instead of assuming a fixed milk volume of 150 mL/kg, which may not accurately represent the dynamic nature of THC exposure in both the mother and infant, we calculated the volume of milk ingested by the baby during each feeding session based on the infant’s age. By combining the breastmilk concentration and breastmilk intake volume, we derived the dose of THC to which the baby is exposed per breastfeeding session. It is important to note that this approach of considering individualized milk intake accounts for some of the variability in THC exposure in infants.

In Figure 11, we present the corresponding infant plasma concentration of THC at each breastfeeding period, highlighting the impact of maternal smoking frequency on infant THC exposure. Notably, even though the virtual subjects were dosed per kilogram of body weight, neonates up to 1 month old, who have the smallest weight, exhibited the highest THC exposure. This is attributed to the incomplete maturation of metabolizing enzymes in neonates. As the baby grows and metabolizing enzyme activity, such as CYP2C9 and CYP3A4, increases, the infants receive higher THC doses due to weight gain, but their plasma THC concentrations remain lower relative to the neonatal period.

#### 3.2.6. Sensitivity Analysis

Sensitivity analysis was specifically conducted on critical model parameters that were fixed to assess their impact on pharmacokinetic exposure. The parameters evaluated included the viscosity and hygroscopicity of THC smoke, the solubility of THC, and the THC breast tissue-to-plasma partition coefficient. The sensitivity tests involved making small changes to each parameter while keeping all other model parameters constant, allowing for rapid identification of parameters that significantly influenced exposure metrics with only slight variations. Notably, the breast tissue-to-plasma partition coefficient exhibited sensitivity values beyond the acceptable limits, indicating that it is an influential input parameter in our lactation PBPK model (see Figure 12).

## 4. Discussion

Limited information is available regarding the concentration of THC in breast milk and its absorption by breastfeeding infants. The American Academy of Pediatrics recommends against cannabis use during lactation. In practice, this leads to variations in recommendations by providers and lactation specialists regarding the safety of breastfeeding for individuals who continue to use cannabis. Although several studies have shown that THC is excreted in breast milk, the actual absorption by infants and subsequent effects have not been evaluated. Given the well-documented benefits of breastmilk, withholding it based on a theoretical risk that has not been fully evaluated is not without its own risks. In our study, we successfully developed a maternal and infant PBPK model, which allows us to describe the disposition of THC in breastmilk and breastfed infant plasma, which is of clinical importance in determining the safety of breastfeeding for those using cannabis.

The application of PBPK models to investigate drug pharmacokinetics in special populations, including lactating women, has garnered significant interest over the years. In a recent research endeavor, PBPK modeling was employed to characterize the breastmilk pharmacokinetics of ten drugs across diverse biopharmaceutics classifications [111]. Specifically, in this study, we developed a PBPK model to estimate THC exposure in human milk and breastfed infants. The lactation PBPK model incorporated a smoke deposition pattern, as well as an oral inhalation compartment, to represent the prevalent routes of cannabis consumption and the likely route for lactating mothers who use cannabis [2].

In traditional approaches, the milk-to-plasma ratio is initially determined empirically, taking into account human milk and drug characteristics. This ratio is then used to calculate the milk concentration by multiplying it by the average plasma concentration [21,112]. Alternatively, a semi-mechanistic approach has also been employed, incorporating clearance between breastmilk and plasma into the equation governing drug transfer between these compartments [58,111]. In our study, we successfully developed a mechanistic lactation PBPK model for THC using a bottom-up approach. Firstly, we separated the breast tissue compartment from the alveoli (milk storage) compartment within the breasts and then modeled THC to transfer either by active secretion or reabsorption directly from the breast tissue plasma or through free permeation from THC within the breast tissue compartment. Subsequently, we reduced our base model into an infant PBPK model based on relevant equations from the literature. Finally, the concentration output from the lactation PBPK was used to drive the dose input for the infant PBPK model. While PBPK models for THC have been developed for pregnant and non-pregnant populations in previous research [43,113,114], this is the first paired PBPK model, to the best of our knowledge, specifically developed for THC in lactating mothers (assuming real-world recreational cannabis use) and infants up to one year old.

In several studies analyzing paired samples of plasma and breastmilk, the observed milk-to-plasma concentration ratio of THC in lactating mothers who smoke marijuana was approximately 7.0 [15,16,17]. However, in our study, we predicted the milk-to-plasma ratio based on concentration to fall between 1.34 and 5.10, while it was 3.3 when determined using the AUC ratio method. The AUC ratio method is considered superior as it considers exposures in both compartments over time, providing a more comprehensive assessment compared to relying on single time-point concentrations, which can vary over time. Furthermore, our lactation PBPK model successfully predicted a THC half-life in breast milk of approximately 39 h, which is consistent with the value reported by Bertrand et al. (27 h) [98]. However, in the study conducted by Wymore et al., a significantly longer half-life of 17 days was reported [16]. This discrepancy in the latter study may be attributed to challenges in ensuring that only mothers who were abstinent from marijuana use over the 42-day period contributed breastmilk samples. Moreover, the plasma analysis of THC in eligible and enrolled mothers who self-reported abstinence indicated that 52% of them had THC present at the beginning of the study. There is currently a lack of information in the literature about THC exposure in infants via breastfeeding. Utilizing our infant-lactation paired PBPK model, we predicted a percent mother-to-infant plasma AUC_(0–24 h)_ of 0.22, 0.18, 0.18, and 0.10%, respectively, for a mother who smoked cannabis once daily and exclusively breastfed a one-month-old infant. Notably, this value increased with an increase in daily cannabis use, showing up to a threefold rise when the daily use increased to six joints. Furthermore, we simulated relative infant doses of 0.59, 0.71, 0.60, and 0.39% for infants up to 1 month, 3 months, 6 months, and 12 months old, respectively. While some studies have reported maternal weight-adjusted relative infant doses of 0.8–8.7%, our model accounted for the dynamics of milk intake per feed, resulting in lower infant exposure to THC through lactation. In our worst-case scenario of maternal cannabis use six times daily during lactation, the maximum infant plasma concentration ranged between 0.084 and 0.167 ng/mL for infants between one month and twelve months, with one-month-old infants showing higher levels. These concentrations are orders of magnitude lower than maternal plasma levels, and the clinical implications of this finding are unclear. Additional data would be needed to interpret THC levels in infants.

THC and its metabolites, including the active compound 11-OH-THC, have consistently been detected in samples from newborns’ umbilical cords and meconium [115,116]. These findings suggest that infants possess the necessary enzyme activity to metabolize THC from birth. However, the analysis of urine and feces from breastfed infants for THC and its metabolites has produced mixed outcomes, revealing varying concentration levels [5]. Nevertheless, assessing THC levels in infant plasma holds greater clinical significance; however, there is a lack of data regarding infants exposed to cannabis through breast milk. Our study predicted THC concentration in infant plasma, although notably lower than previously reported values. This discrepancy is due to prior studies involving instances of accidental cannabis intoxication by infants, resulting in reported infant plasma THC and 11-OH-THC concentrations ranging from 23.9 ng/mL to 54.8 ng/mL and 11.8 ng/mL to 35.1 ng/mL, respectively [117,118]. Our current study exclusively focused on estimating THC concentration, although the intermediate metabolite of THC (11-OH-THC) exhibits similar pharmacological effects to THC.

In a pharmacodynamic investigation of oral cannabis use in healthy adults, self-reported effects, as well as cognitive and psychomotor activity, were more pronounced at plasma THC concentrations of 2.2 ng/mL or higher [119]. Additionally, another review indicated that blood THC concentrations between 2 and 5 ng/mL can lead to substantial driving impairment [120]. Given the doses considered via maternal cannabis smoking and subsequent infant dose through breastfeeding, the simulated infant plasma concentrations (ranging from 0.084 to 0.167 ng/mL) were lower than the threshold for pharmacodynamic effects in adults, which is reassuring. However, the potential health risks or impairments to the developing baby at these concentrations remain uncertain. More research and larger-scale studies are needed to validate these findings and provide further insights into the potential effects of THC exposure during lactation. As the prevalence of cannabis use among lactating mothers continues to rise, it is essential to prioritize research in this area to ensure the safety and well-being of both mothers and their breastfed infants.

The development of PBPK models for special populations, including our lactation and infant PBPK model for THC, is usually confronted with a notable challenge—the limited availability of clinical data to adequately evaluate model performance. In the case of our lactation model, we relied on breastmilk data from the only study that reported relevant dosing information, including the amount, smoking pattern, and breastmilk sample collection time, to verify the accuracy of our model predictions. For our infant model, we scaled an adult population pharmacokinetic model down to infants and compared our model predictions with this, and we did not account for THC metabolites. Notwithstanding these limitations, our model predictions provided valuable insights and represent a significant step forward in enhancing our understanding of THC pharmacokinetics in lactating women and their breastfed infants. This information is crucial for assessing the potential risks and safety implications of cannabis use during lactation and guiding appropriate recommendations for marijuana use in lactating women.

## 5. Conclusions

In this study, a PBPK model was developed to describe the pharmacokinetics of THC in the breast milk of lactating cannabis users and its potential exposure to breastfeeding infants. While the model predicted higher THC concentrations in breast milk, the corresponding exposure in infant plasma was significantly lower. The clinical implications of this lower exposure remain uncertain, underscoring the need for further research to determine whether this level of exposure poses any potential harm to infants.

## Figures and Tables

**Figure 1 pharmaceutics-15-02467-f001:**
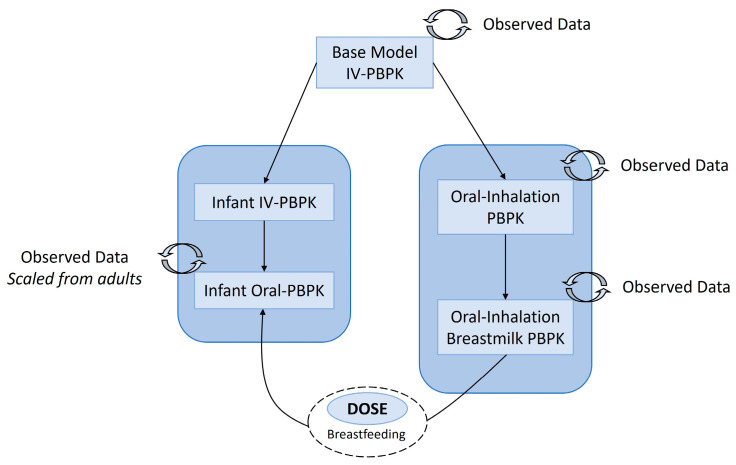
Workflow of model development for the maternal lactation and infant PBPK model. IV, intravenous; PBPK, physiologically based pharmacokinetic model.

**Figure 2 pharmaceutics-15-02467-f002:**
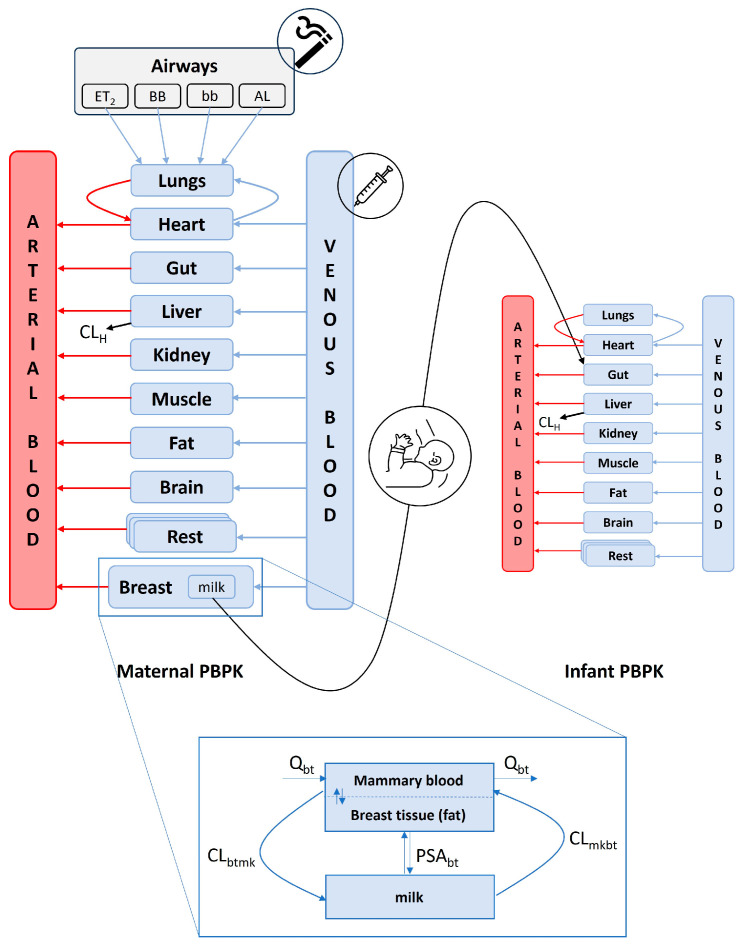
Schematics of the maternal and infant PBPK model structure, highlighting modes of drug input, distribution, and elimination. ET_2_, extrathoracic (excluding the nose); BB, bronchial; bb, bronchiolar; and AL, alveolar region; CL_H_, hepatic clearance; Q_bt_, blood flow to the breast; CL_btmk_, drug clearance from the breast into milk; CL_mkbt_, drug clearance from milk into the breast.

**Figure 3 pharmaceutics-15-02467-f003:**
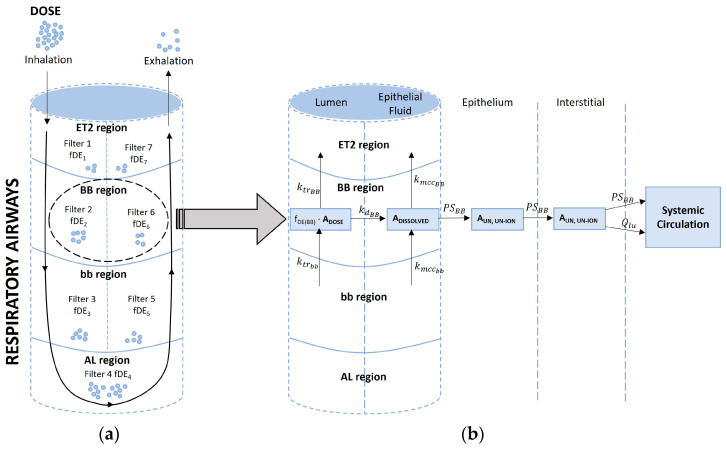
Schematics of particle deposition following (**a**) inhalation and exhalation and (**b**) absorption of deposited particles through the bronchial region. fDE, fraction deposited; $k_{{tr}_{BB}}$, $k_{{tr}_{bb}}$, transit rate constant at bronchial and bronchiolar regions (1/h), respectively; $k_{d_{BB}}$, dissolution rate constant at bronchial region (1/h); $k_{{mcc}_{BB}}$, $k_{{mcc}_{BB}}$, mucociliary transit rate constant at bronchial and bronchiolar regions (1/h), respectively; ${PS}_{BB}$, permeability surface area product of bronchial epithelium (L/h); $Q_{lu}$, blood flow to the lungs.

**Figure 4 pharmaceutics-15-02467-f004:**
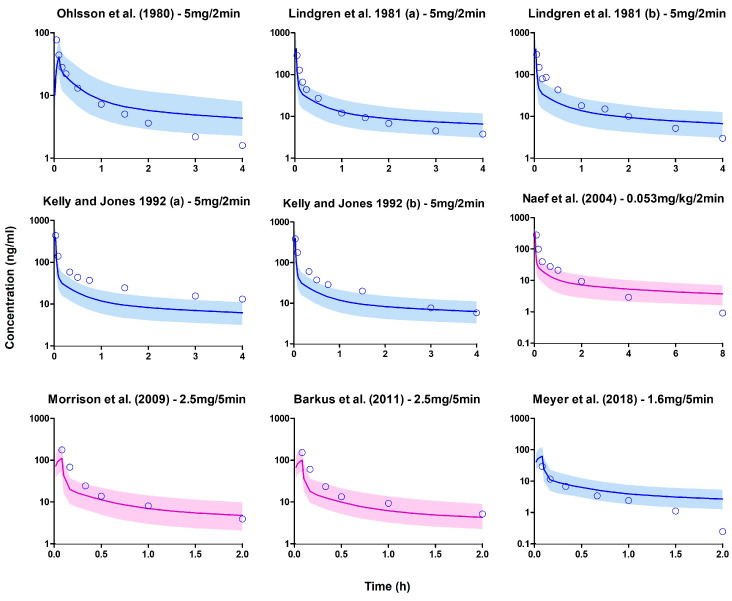
Plasma concentration profiles following a short intravenous infusion. The open blue circle represents the observed data. The solid line depicts the median PBPK predicted concentrations, while the shaded area (5th to 95th percentile) indicates the 90% prediction interval. The purple and blue profiles display the overlay of PBPK simulated values with the observed values in the training and verification datasets, respectively [85,86,87,88,89,90,91].

**Figure 5 pharmaceutics-15-02467-f005:**
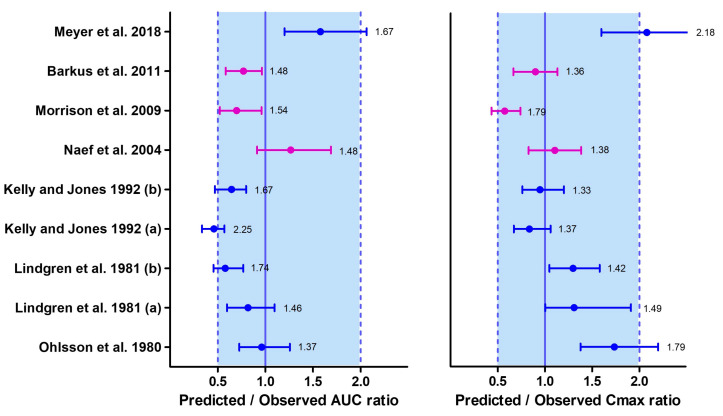
Median (interquartile range) predicted/observed ratios for AUC and Cmax following intravenous administration of THC. The purple median (IQR) corresponds to the results from the training datasets, while the blue median (IQR) represents the results from the verification datasets. Each median (IQR) on the plot is accompanied by the absolute average fold errors (AAFE), which provide an overall measure of the deviation between simulated and observed values. The acceptance criterion for the ratio is set between 0.5 and 1, while the criterion for the AAFE is set between 1 and 2. Multiple datasets were included from certain studies where both chronic (a) and casual (b) cannabis use scenarios were investigated within the same study [85,86,87,88,89,90,91].

**Figure 6 pharmaceutics-15-02467-f006:**
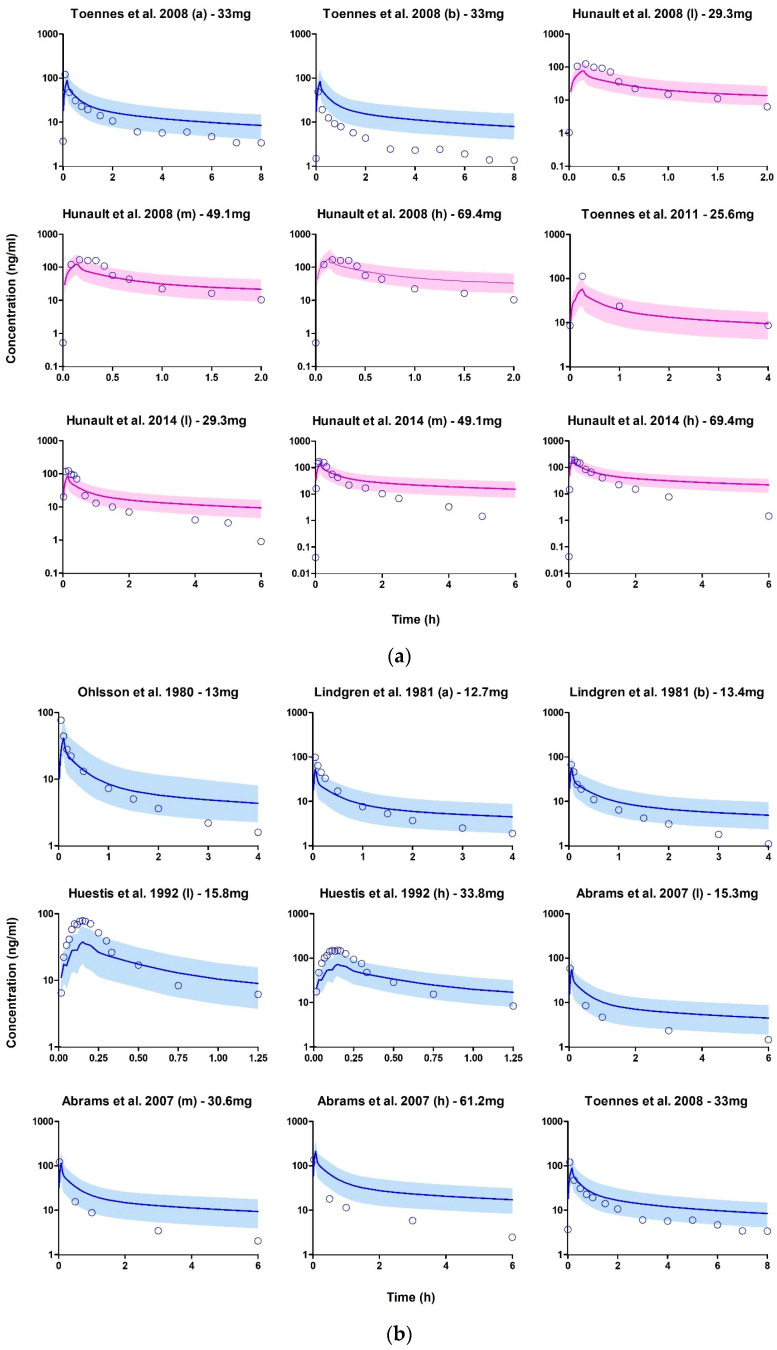
(**a**) Plasma concentration profiles following THC inhalation after smoking cannabis. The open blue circle represents the observed data. The solid line depicts the median PBPK predicted concentrations, while the shaded area (5th to 95th percentile) indicates the 90% prediction interval. The purple and blue profiles display the overlay of PBPK simulated values with the observed values in the training and verification datasets, respectively. (**b**) Plasma concentration profiles following THC inhalation after smoking cannabis. The open blue circle represents the observed data. The solid line depicts the median PBPK predicted concentrations, while the shaded area (5th to 95th percentile) indicates the 90% prediction interval. The blue profile displays the overlay of PBPK simulated values with the observed values in the verification datasets [85,86,92,93,94,95,96,97].

**Figure 7 pharmaceutics-15-02467-f007:**
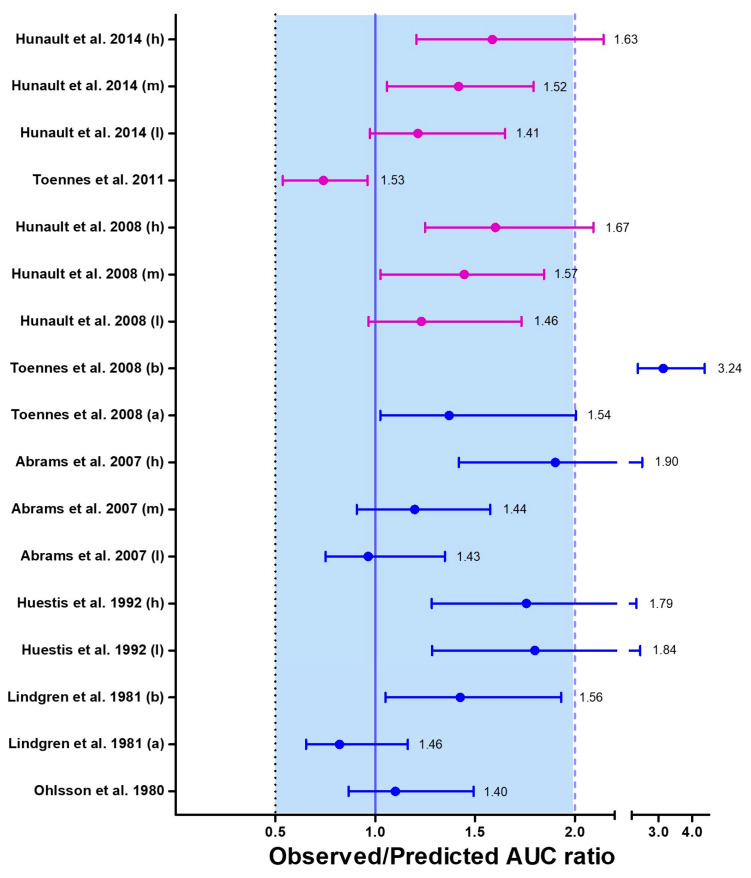
Median (interquartile range) predicted/observed ratios for AUC following inhalation of cannabis smoke. The purple median (IQR) corresponds to the results from the training datasets, while the blue median (IQR) represents the results from the verification datasets. Each median (IQR) on the plot is accompanied by the absolute average fold errors (AAFE). The acceptance criterion for the ratio is set between 0.5 and 1, while the criterion for the AAFE is set between 1 and 2. Multiple datasets were included from certain studies that examined a range of scenarios, including chronic (a) and casual (b) cannabis use, as well as variations in THC content such as low (l), medium (m), and high (h) THC content in cannabis joints within the same study [85,86,92,93,94,95,96,97].

**Figure 8 pharmaceutics-15-02467-f008:**
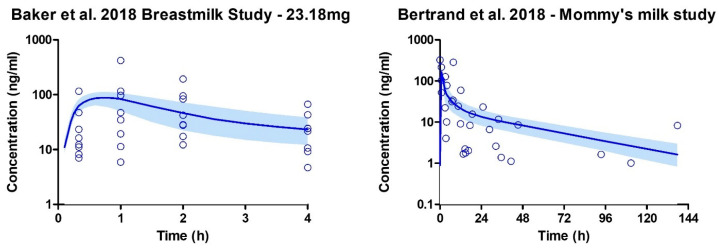
Breastmilk concentration profiles following THC inhalation after smoking cannabis. The open blue circle represents the observed data. The solid line depicts the median PBPK predicted concentrations, while the shaded area (5th to 95th percentile) indicates the 90% prediction interval. Eight lactating mothers participated in the Baker et al. study, and their profile is shown on the graph. However, in the Bertrand et al. study, 50 mothers who reported recent marijuana use contributed one breastmilk sample at different times [13,98].

**Figure 9 pharmaceutics-15-02467-f009:**
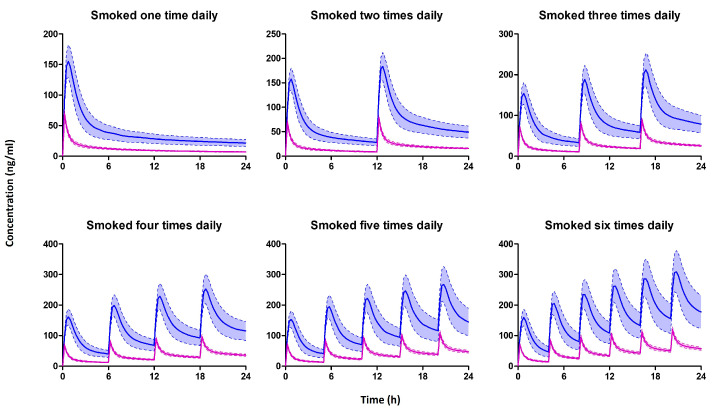
The predicted median (IQR) plasma (Purple) and breastmilk (blue) concentration of mothers who smoked one, two, three, four, five, or six times per day. Multiple smoking sessions were spaced evenly within a 24 h period.

**Figure 10 pharmaceutics-15-02467-f010:**
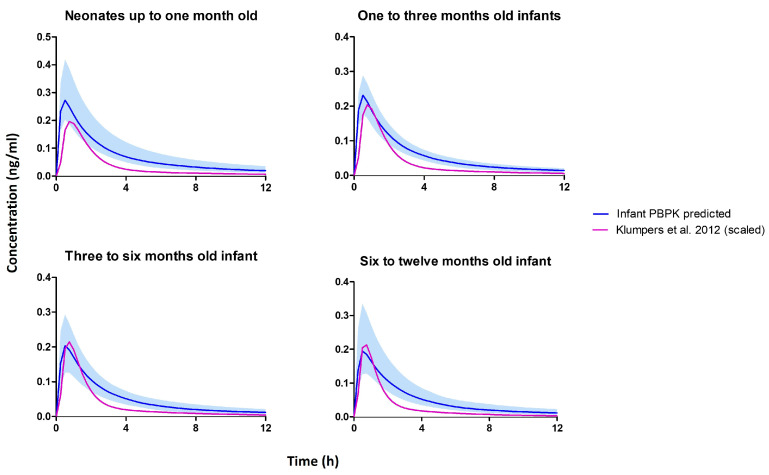
Simulated plasma concentration profile from the infant PBPK model and scaled Klumpers et al. [108] population pharmacokinetic model.

**Figure 11 pharmaceutics-15-02467-f011:**
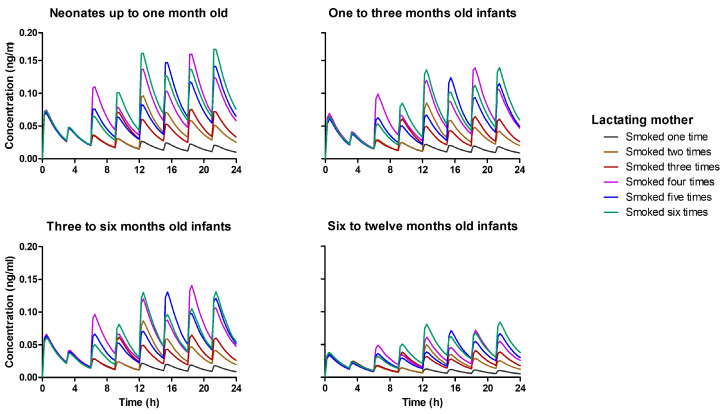
Simulated plasma concentration profile for infants up to one year of age. It was assumed baby feeds every three hours, and the breast milk concentration at the time of feeding depends on the number of times the lactating mother smoked. One to six times smoking frequency per day was tested and represented by different colors, as shown in the legend.

**Figure 12 pharmaceutics-15-02467-f012:**
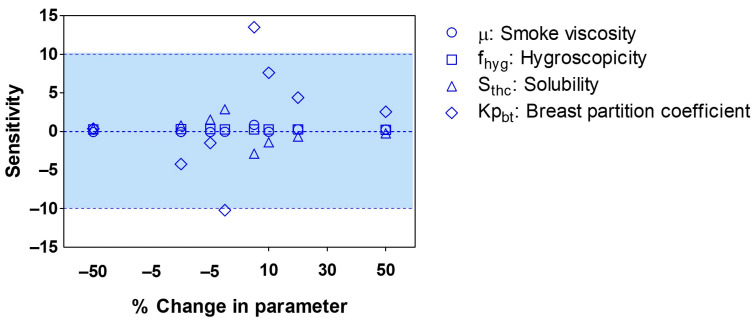
Sensitivity analysis of exposure metrics to variations in fixed model input parameters. The viscosity and hygroscopicity of THC smoke were assessed in relation to the fraction deposited, while solubility was evaluated against the plasma area under the curve (AUC). Additionally, the breast partition coefficient was examined in correlation with breast milk AUC.

**Table 1 pharmaceutics-15-02467-t001:** System-specific physiological parameters for the PBPK model.

Organ	Weight (g) ^a^	ρ	Q ^b^	fVwt	fVnl	fVph	fVew	fViw	Reference
LBM	0.252⋅Weight+0.473⋅Height−48.3								[25]
Adipose	$17.4+0.65\cdot Age-0.01\cdot {Age}^{2}+9\times 10^{-5}\cdot {Age}^{3}$	0.9196	0.085	0.286	0.609	0.005	0.135	0.017	[26,27,28,29,30,31]
Bone	0.21⋅LBM	1.176	0.05	0.45	0.074	0.0011	0.1	0.346	[26,31,32]
Brain	$653+95.4\cdot Age-4.32\cdot Age^{2}+0.0729\cdot Age^{3}-0.000413\cdot Age^{4}$	1.04	0.12	0.78	0.051	0.0565	0.162	0.62	[26,31]
Gut	1100	1.045	0.17	0.76	0.0487	0.0163	0.282	0.475	[26,31]
Muscle	0.29⋅Weight	1.06	0.12	0.71	0.022	0.0072	0.118	0.63	[26,27,31]
Heart	$25.39+15.70\cdot Age-0.3603\cdot Age^{2}+0.004\cdot Age^{3}-1.75\times 10^{-5}\cdot Age^{4}$	1.055	0.05	0.78	0.0115	0.0166	0.32	0.456	[26,31]
Spleen	$8.99+10.13\cdot Age-0.24\cdot Age^{2}+0.0018\cdot Age^{3}-2.95\times 10^{-6}\cdot Age^{4}$	1.06	0.03	0.79	0.0201	0.0198	0.207	0.579	[26,31]
Kidney	$20.4+18.7\cdot Age-0.511\cdot Age^{2}+0.0054\cdot Age^{3}-1.88\times 10^{-5}\cdot Age^{4}$	1.035	0.17	0.76	0.0207	0.0162	0.273	0.483	[26,31]
Lungs	$115+36.8\cdot Age-0.4\cdot Age^{2}$	1.05	1.0	0.78	0.003	0.009	0.336	0.446	[26,31]
Liver	$145+104\cdot Age-3.2\cdot Age^{2}+0.043\cdot Age^{3}-0.0002134\cdot Age^{4}$	1.054	0.27	0.73	0.0348	0.0252	0.161	0.573	[26,31]
Rest	$Weight-\sum W_{tissues}$		0.065	0.70 ^ca^	0.02 ^c^	0.01 ^c^	0.652	1.581	[31]
Blood	$0.36\cdot Height^{3}+0.03\cdot Weight+0.1833$								[33,34]
Plasma	$0.6\cdot W_{Blood}$			0.95	0.0032	0.0021			[34]
Venous	$0.7\cdot W_{Blood}$								[35]
Artery	$W_{Blood}-W_{Venous}$								

^a^ Equations were either reported or generated from the digitized plot; ^b^ values were reported as fractions; ^c^ Values were assumed. ρ, specific gravity (g/cm^3^); Q, organ blood flow; fV, fractional volume; wt, nl, ph, ew, iw, water, neutral lipids, extracellular water, and intracellular water, respectively; $W_{tissues}$, $W_{Blood}$, $W_{Venous}$ weight of tissues, blood, and venous blood, respectively. The volume of each organ was calculated by multiplying the organ weight and specific gravity.

**Table 2 pharmaceutics-15-02467-t002:** Physicochemical and biochemical parameters of THC used in the PBPK model.

Parameter	Definition	Value	Reference
Dose (mg)	Standard joint weighing 0.32 g containing 14.14% THC	45.2	[37,38]
MW (g)	Molecular weight (C_21_H_30_O_2_)	314.5	[39]
Log P	Octanol-water partition coefficient	6.97	[40]
BP	Blood-to-plasma ratio	0.667	[41]
pKa	Dissociation constant	10.6	[42]
fu_p_	Unbound fraction in plasma	0.0022	[43]
fu_mic_	Unbound fraction in liver microsomes	0.04	[44]
Vmax_CYP2C9_ (pmol/min/mg)	CYP2C9 maximum reaction rate	624	[44]
Km_CYP2C9_ (μmol/L)	CYP2C9 concentration at half maximum reaction rate	0.07	[44]
Vmax_CYP3A4_ (pmol/min/mg)	CYP3A4 maximum reaction rate	4905	[44]
Km_CYP3A4_ (μmol/L)	CYP3A4 concentration at half maximum reaction rate	5.48	[44]
Vmax_PGP,BR_ (μmol/h)	PGP brain maximum reaction rate	0.0123	[45]
Km_PGP,BR_ (μmol/L)	PGP brain concentration at half maximum reaction rate	49.1	[45]
S_THC_ (mg/L)	THC Solubility	2.8	[39]
PSA (Å^2^)	THC Polar Surface Area	29.5	[39]
HBD	THC Hydrogen Bond Donor	1	[39]
d_ae_ (μm)	Aerodynamic diameter of THC smoke particle	0.39	[46]
ρ (g/cm^3^)	Density of THC smoke particle	3	[26]
χ	Dynamic shape factor of smoke particle	1.5	[26]
f_hyg_	Hygroscopic growth rate factor	1.5 ^a^	[47]

^a^ Assumed based on the hygroscopic growth of particles from a Kentucky 3R4F reference cigarette. CYP, Cytochrome P450; PGP, P-glycoprotein; THC, Tetrahydrocannabinol.

**Table 3 pharmaceutics-15-02467-t003:** Characteristics of the model verification and validation dataset.

Dose	Study Description	Subjects	Age (year)	Weight (kg)	Purpose	Reference
Intravenous						
5 mg/2 min	RD, XO	11 (100% male)	18–35	-	Verification	[85]
5 mg/2 min	PC, XO	9 (89% male)	29.2 (5.2)	73.7 (10.3)	Verification	[86]
5 mg/2 min	PC, XO	9 (89% male)	25.3 (4.9)	68.3 (9.6)	Verification	[86]
5 mg/2 min	-	8 (100% male)	24–45	64–87	Verification	[87]
0.053 mg/kg/2 min	RD, DB, XO	8 (50% male)	26–35	60 (8)–80 (5)	Training	[88]
2.5 mg/5 min	DB, PC	22 (100% male)	28 (6)	-	Training	[89]
2.5 mg/5 min	RD, DB, PC	11 (100% male)	26.3 (4.2)	-	Training	[90]
1.6 mg/5 min	P1, SC, OL, 2-periods	11 (55% male)	18–40	74	Verification	[91]
Inhalation (Smoking) *						
13 mg/6 min	RD, XO	11 (100% male)	18–35	-	Verification	[85]
12.7 mg/3 min	PC, XO	9 (89% male)	29.2 (5.2)	73.7 (10.3)	Verification	[86]
13.4 mg/3 min	PC, XO	9 (89% male)	25.3 (4.9)	68.3 (9.6)	Verification	[86]
15.8 mg, 33.8 mg/11.2 min	RD, DB, LS	6 (100% male)	31.3 (29–36)	77.6 (65–93)	Verification	[92]
15.3 mg, 30.6 mg, 61.2 mg/6 puffs	RD, CT, PS	18 (83% male)	21–45	-	Verification	[93]
33 mg/10 min	Two-way, DB, PC	12 (67% male)	22 (20–31)	66 (55–84)	Verification	[94]
29.3 mg, 49.1 mg, 69.4 mg/22 min	Four-way, RD, DB, PC, XO	24 (100% male)	24 (4)	74 (5)	Training	[95]
25.6 mg/15 min	DB, XO, PC	19 (74% male)	23 (19–38)	61.5	Training	[96]
29.3 mg, 49.1 mg, 69.4 mg/22 min	RD, DB, PC, XO	24 (100% male)	24 (4)	-	Training	[97]
Smoking and lactation *						
23.18 mg/15 min	Pilot 2–5 mo Postpartum PK study	8 (100% female)	18–45	-	Training and verification	[13]
45.25 mg/8.7 min	Mommy’s milk study	50 (100% female)	22–41	-	verification	[98]

* The standard joint cigarette content and/or smoking topography was assumed where there is a lack of information in the study. RD, randomized; XO, crossover; PC, placebo-controlled; CT, controlled; DB, double-blind; LS, Latin-square; PS, pilot study; P1, phase one; SC, single center; OL, open-label. Age and weight were reported as range or mean (standard deviation).

**Table 4 pharmaceutics-15-02467-t004:** Comparison of observed and predicted breastmilk exposure metrics.

	Median (Range)			
Parameter (Units)	Observed	Predicted	Ratio	AAFE
AUC (ng⋅h/mL)	110.5 (33.9–744.4)	194 (99.2–301.8)	1.76	1.67
C_avg_ (ng/mL)	27.6 (8.4–186.1)	48.5 (24.8–75.5)	1.76	1.67
C_max_ (ng/mL)	44.7 (12.2–420.3)	88.7 (55.6–121.1)	1.98	1.93
T_max_ (h)	1 (1–2)	0.8	0.8	
Infant dose (mcg/kg/d)	4.1 (1.3–27.9)	7.3 (3.7–11.3)	1.78	1.68
RID (%)	1.3 (0.4–8.7)	2.2 (1.1–3.4)	0.59	1.61

C_avg_, average concentration; C_max_, maximum concentration; T_max_, time to maximum concentration; RID, relative infant dose in percentage; AAFE, absolute average fold error.

**Table 5 pharmaceutics-15-02467-t005:** Predicted C_max_, AUC, and the milk-to-plasma ratio of THC in lactating mothers.

	Breastmilk			Plasma			Infant AUC_(0–24 h)_ (RID)			
Joints	C_max_	AUC_(0–24 h)_	C_avg_	C_max_	AUC_(0–24 h)_	MP Ratio	1 mo	2 mo	6 mo	12 mo
/day	ng/mL	ng·hr/mL	ng/mL	ng/mL	ng·hr/mL		ng·hr/mL (%)	ng·hr/mL (%)	ng·hr/mL (%)	ng·hr/mL (%)
1	155	924.9	38.5	69.9	273.4	3.4	0.59 (0.74)	0.49 (0.88)	0.49 (0.74)	0.26 (0.48)
2	184	1554	64.8	78.6	444.6	3.5	0.98 (0.63)	0.81 (0.74)	0.80 (0.63)	0.48 (0.41)
3	212	2144	89.3	89.1	636.4	3.4	1.02 (0.57)	0.84 (0.68)	0.84 (0.57)	0.54 (0.37)
4	253	2876	119.8	97.5	806.5	3.6	1.75 (0.58)	1.49 (0.67)	1.49 (0.58)	0.77 (0.38)
5	268	3223	134.3	110	968.5	3.3	1.57 (0.52)	1.26 (0.62)	1.32 (0.52)	0.74 (0.34)
6	309	3996	166.5	120	1181	3.4	1.85 (0.54)	1.50 (0.64)	1.41 (0.54)	0.92 (0.35)

C_max_, maximum concentration; AUC_(0–24 h)_; Area-under-the-curve from 0 to 24 h; C_avg_, average concentration; MP, mother-to-plasma; RID, relative infant dose in percentage.

## Data Availability

No new data were created or analyzed in this study.

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
