# Peer review of "Predicting Maternal and Infant Tetrahydrocannabinol Exposure in Lactating Cannabis Users: A Physiologically Based Pharmacokinetic Modeling Approach"

_pharmaceutics, 2023, doi:10.3390/pharmaceutics15102467_

Round 1
Reviewer 1 Report
Comments
Pharmaceutics: 2627036
Date: 16th September 2023
Title: Cannabis and Lactation: A Physiologically Based Pharmacokinetic Model of THC Exposure in Breastmilk and Infant Plasma in Lactating Cannabis Users.
The research article is suitably designed, informative, well-written and fit for the Journal scope. I found that Shenkoya et al. highlighted the concept of simulation using PBPK model for cannabis receiving breast feeding mother. The work is comprehensive and suitable for publications after major revision.  Comments    
- Abstract must be rewritten with major findings (values) and coherent. The selected title is too long and keywords should be properly selected to give outline of the abstract. The sentence “The final PBPK model included oral inhalation and breastmilk compartments and was scaled to infants by incorporating age-related ontogeny equations.” must be rewritten to avoid confusion. In the abstract section, I recommend to mention the simulation time point while running the program. Moreover, body weight of selected breastfeeding mother and apparent permeability coefficient value fed into the respective tab of the program.
- I recommend to present a table of input parameters for THC such as dose, molecular weight, empirical formula, reference solubility, dose, dosing volume, dosing interval, apparent permeability, particle size, dose number, bioavailability, and pharmacokinetic parameters (Cmax, Tmax, AUC, Vd, steady state concentration after oral administration, and dosage form selected in the tab). I found several abbreviations without its expanded form. PK parameters should be defined in bracket to understand the terms. Similarly, authors could not maintain consistency in unit “ml or mL”.
- The model uses various input parameters (experimental, by-default, and literature) and therefore, it must be elaborated in the introduction section. Validation parameters such as “fold error” must mathematically defined in the manuscript.
- Figure 9 and figure 10 need to be merged together to reduce the number of figures. In figure 10, why did authors simulate up to 12 h whereas it was for 24 h in figure 9?
- I suggest to present figure 12 in 3-dimensional plot to understand the parameter sensitivity analysis report. What was the purpose of selection of these factors?
- Fold error is a statistical value to validate model. I recommend to present a suitable mathematical equation for AAFE (RSC Adv., 2016,6, 93147-93161 and https://doi.org/10.1016/j.ijpharm.2019.05.061). FE must be lower than 2 as acceptance criteria.
- I recommend to include a short conclusion section after discussion. It might be good to understand the simulated outcomes to readers.
- Authors have used repeatedly “physiological based PK” in the manuscript. Once abbreviated, it must be followed in the subsequent sentences.
- The sentence “In a pharmacodynamic dose-effect study of oral cannabis ingestion in healthy adults, plasma concentrations of THC equal to or greater than 2.2 ng/ml resulted in pronounced subjective effects and significantly increased cognitive and psychomotor activity” must be rewritten to avoid plagiarism. Section 3.2.1, line 417 -419 must be rewritten to avoid plagiarism.
Minor language correction is required.
Reviewer 2 Report
Authors of the presented manuscript evaluated their adopted physiologically based pharmacokinetic model for characterizing the ADME profile of THC within mother breastmilk and plasma of breastfeeding infants following cannabis smoking by lactating mothers. Findings are preceding the current evidence within the literature and provide valuable clinical data in term of risk and safety that would guide appropriate recommendations for marijuana use within lactating mothers. The manuscript is highly valuable within its field and the rational is well-represented. Minor suggestions are addressed as following:
1. In Figures 5 and 7, authors are advised to perform pooling of weighed data to obtain an overall summary from the identified studies.
2. Heterogeneity test could be adopted to assess and detect any statistically-relevant heterogeneity or variability between the estimates of connection regarding the identified relevant studies.
3. Flow chart describing the adopted study selection process could be presented, including descriptions for the identification, screening, eligibility, and inclusion steps.
Minor editing of English language required.
Reviewer 3 Report
The authors are commended for tackling this important question on a complex drug with very few data. I take some exception with some model details and have some comments below.
Line 56. High protein binding is most definitely NOT a favorable property for passage into milk. Quite the contrary.
Line 74. Sentence seems incomplete
Lines 127-128. The description of the breast and milk compartment needs more details. Is it a direct transfer from maternal blood into milk or does it go from maternal blood to maternal breast tissue to milk, i.e., a one-compartment or two-compartment model? From eqs. 14 and 15, it looks like a two-compartment model. A separate detailed graphic may be in order.
Lines 150-154 need proper alignment. Are they part of the previous paragraph or something new?
Lines 170-171. Sentence is split
Line 258. Figure 2 does not clearly show the details of breast to milk transfer (see comment on lines 127-128).
Line 264. The terms “secretion” and “reuptake” imply active processes. Is there any evidence for active processes? If not, “excretion” and “reabsorption” might be better terms.
Drugs in the plasma partition out of the bloodstream and into the interstitial fluid and then across the epithelial cells and into milk. Oftentimes, the interstitial fluid transfer is fast enough to ignore. What are the authors defining as “breast tissue”? Drugs do not partition into breast stroma or adipose tissue on the way to the milk. Clearance from milk to breast does not seem to fit into the model. Milk is cleared by back-diffusion into the plasma by the reverse of the above. I would consider partitioning into the fat in the breast part of the overall fat compartment.
Line 315. Change “pediatrics” to “children”. Pediatrics is a department.
Line 318. Changing pediatric to infant would be good here.
Lines 379-80. Once daily is a chronic user. The true casual smoker is not accounted for. For example, once/week. This is important for clinicians trying to answer patient queries about casual use. A simulation of once/week use would be helpful.
Line 538. Maybe this is correct for Lithuania, but it certainly does not match the breastfeeding durations or exclusivity in the US! See https://www.cdc.gov/breastfeeding/data/reportcard.htm
Lines 578-584. The authors are also assuming exclusive breastfeeding, which is not the case for many women under 6 months of age. See the CDC report card. This should be mentioned. Also, exclusive breastfeeding is not recommended after 6 months of age as seems to be assumed in Fig 11, 6-12 months of age.
Lines 606-608. This high sensitivity may be a result of model misspecification as noted above.
Line 622. I would change “purported” to “well documented”. See Meek JY, Noble L. Technical report: Breastfeeding and the use of human milk. Pediatrics. 2022;150:e2022057989; doi: 10.1542/peds.2022-057989
Line 650. It would be more “real-world” if some less frequent use (1x/week) modeling were added.
Line 670. Insert “exclusively” before “breastfed”
Lines 673-679. The 6-12 months assumes exclusive breastfeeding.
Line 695. It may be that brain (a fatty tissue) levels are more important than plasma levels.
Round 2
Reviewer 1 Report
The authors have revised the manuscript carefully. I recommend for publication.
Minor correction required